# Oncolytic Vaccinia Virus Carrying *Aphrocallistes vastus* Lectin (oncoVV-AVL) Enhances Inflammatory Response in Hepatocellular Carcinoma Cells

**DOI:** 10.3390/md20110667

**Published:** 2022-10-26

**Authors:** Xinnan Zheng, Weizhe Xu, Qi Ying, Jiajun Ni, Xiaoyuan Jia, Yanrong Zhou, Ting Ye, Gongchu Li, Kan Chen

**Affiliations:** College of Life Sciences and Medicine, Zhejiang Sci-Tch University, Hangzhou 310018, China

**Keywords:** *Aphrocallistes vastus* lectin, oncolytic vaccinia virus, AP-1, hepatocellular carcinoma

## Abstract

*Aphrocallistes vastus* lectin (AVL) is a C-type marine lectin derived from sponges. Our previous study demonstrated that oncolytic vaccinia virus carrying AVL (oncoVV-AVL) significantly enhanced the cytotoxicity of oncoVV in cervical cancer, colorectal cancer and hepatocellular carcinoma through the activation of Ras/ERK, MAPK/ERK and PI3K/Akt signaling pathways. In this study, the inflammatory response induced by oncoVV-AVL in a hepatocellular carcinoma cell (HCC) model was investigated. The results showed that oncoVV-AVL increased the levels of inflammatory cytokines including IL-6, IL-8 and TNF-α through activating the AP-1 signaling pathway in HCC. This study provides novel insights into the utilization of lectin AVL in the field of cancer therapy.

## 1. Introduction

Oncolytic viruses (OVs) are defined as genetically engineered or naturally occurring viruses that selectively replicate in and kill cancer cells without harming healthy cells. OVs have been identified as a promising therapeutic option for a variety of cancers [1]. The approvals of talimogene laherparepvec (T-VEC) [2] and a modified herpes simplex virus (G47Δ) [3] further increases the research interest of OVs. 

Recently, the roles of marine organism-derived compounds have been widely reviewed. Studies have suggested that marine organisms may be an inspiring tool to develop new anticancer agents [4]. Lectins are widely distributed in marine bioresources including marine cyanobacteria, algae, invertebrate animals and fish [5,6,7]. *Aphrocallistes vastus* lectin (AVL) is a Ca^2+^-dependent C-type lectin produced by sponges [8]. Our previous studies demonstrated that *Aphrocallistes vastus* lectin (AVL) could significantly enhance the anti-tumor effect of oncolytic vaccinia virus (oncoVV). In colorectal cancer, AVL enhanced replication of oncoVV by activating MAPK/ERK signaling pathways [9]. In cervical cancer, oncoVV-AVL interfered with Raf/ERK signaling pathways and inhibited IFN induction, thereby enhancing viral replication [10]. In hepatocellular carcinoma, oncoVV-AVL promoted viral replication through multiple signaling pathways including PI3K/Akt, Hippo and MAPK/ERK [11].

Currently, basic research in immunology and virology has opened up a new avenue for OV therapy. However, the treatment of cold tumors characterized by a lack of T cell infiltration is a great challenge, because adaptive immune response is not maintained in cold tumors. As the results, the new focus is on targeting the immune response, rather than the tumor itself [12]. A series of studies have shown that vaccinia virus can not only directly lyse tumor cells [13,14], but also initiate the anti-tumor immune responses [15,16,17]. On the other hand, several studies indicated that C-type lectin could enhance the immune response [18,19]. 

Our previous investigation illustrated that both IFN-α and IFN-β expressions were enhanced via the infection of oncoVV-AVL [9,10,11], indicated that oncoVV-AVL stimulated the anti-tumor immune response in hepatocellular carcinoma. Therefore, in this research, we focus on the association of oncoVV-AVL infection and the cellular immune response in hepatocellular carcinoma cells.

## 2. Results

### 2.1. OncoVV-AVL Promoted the Transcription of Inflammatory Cytokines

To investigate the association between inflammatory cytokines production and oncoVV-AVL infection, the total RNA of cells was extracted, and the mRNA levels of IL-6, IL-8 and TNF-α were assessed by qRT-PCR assay (Figure 1a–d). The results illustrated that the transcription levels of IL-6, IL-8 and TNF-α in the oncoVV-AVL treatments were significantly increased as compared with PBS and oncoVV treatments in PLC/PRF/5, Hep3B, SK-HEP-1 and HuH-7 cells. In particular, TNF-α was increased more than 40-fold in PLC/PRF/5 cells, and IL-6 was increased more than 20-fold in Hep3B cells. In SK-HEP-1 and HuH7 cells, IL-8 was increased by 90-fold and 30-fold, respectively. This indicated that AVL greatly enhanced oncoVV-mediated transcription of inflammatory cytokines.

### 2.2. OncoVV-AVL Promoted the Production of Inflammatory Cytokines

In order to confirm the changes of these inflammatory cytokines affected by oncoVV or oncoVV-AVL, ELISA assay was performed to detect the production of the cytokines. The results were shown in Figure 2a–c. The production of IL-6 showed no significant difference between the oncoVV-AVL treatments and the oncoVV treatments in PLC/PRF/5 cells, but in Hep3B cells, the production of IL-6 was significantly increased in the oncoVV-AVL treatments, which was approximately 8-fold higher than that in PBS treatments and oncoVV treatments. The production of IL-8 was significantly increased in oncoVV-AVL treatments as compared with PBS and oncoVV treatments in PLC/PRF/5 cells, but not in Hep3B cells. TNF-α was also significantly increased in PLC/PRF/5 and Hep3B cells by oncoVV-AVL treatments as compared with PBS treatments and oncoVV treatments. These results further demonstrated that oncoVV-AVL infection initiated the production of these inflammatory cytokines in HCC.

### 2.3. oncoVV-AVL Promoted the Transcriptional Activity of AP-1 in HCC

To explore the activation of AP-1 and NF-κB, as well as the inflammatory response induced by oncoVV-AVL, we investigated the transcriptional activities of transcription factors AP-1 and NF-κB using dual-luciferase reporter gene assay. The results were shown in Figure 3a,b. In all cells, especially in PLC/PRF/5 and Hep3B cells, the transcriptional activity of AP-1 was significantly enhanced by oncoVV-AVL infection. However, oncoVV-AVL did not significantly activate NF-κB. This indicated that AP-1 was involved in the inflammatory response caused by oncoVV-AVL infection.

To investigate the activation of AP-1, we tested the expression and phosphorylation of c-Fos and c-Jun at protein levels. As shown in Figure 4, in PLC/PRF/5 cells, oncoVV induced the expression and phosphorylation of c-Fos, and oncoVV-AVL further increased the phosphorylated levels of c-Fos. Simultaneously, oncoVV-AVL not only enhanced the protein levels of c-Jun, but also stimulated the phosphorylation of c-Jun. In Hep3B cells, both of the expressions of c-Fos and p-c-Fos were induced by oncoVV, and they were further increased by oncoVV-AVL. In addition, oncoVV-AVL not only restored and further increased the protein levels of c-Jun which was inhibited by oncoVV, but also significantly stimulated the phosphorylation of c-Jun (Figure 5). In SK-HEP-1 cells, the expressions of c-Fos, p-c-Fos and p-c-Jun were induced by oncoVV-AVL, but not by oncoVV. OncoVV-AVL restored and further significantly enhanced the expressing level of c-Jun protein which was inhibited by oncoVV (Figure 6). In HuH-7 cells, both oncoVV and oncoVV-AVL increased the expression of c-Fos, c-Jun and p-c-Jun, while oncoVV-AVL stimulated the expression of p-c-Fos (Figure 7).

To summarize, our results indicated that oncoVV-AVL induced the activity of AP-1 through promoting the expression and phosphorylation of c-Fos and c-Jun, the two compositing proteins of AP-1, which initiated the inflammatory response. 

## 3. Discussion

The progression of cancer is critically regulated by the microenvironment, which may provide factors to promote cancer development or escape from the surveillance of the host immune system [20]. The immune cells within the cancer microenvironment include tumor-associated macrophages, myeloid-derived suppressor cells, regulatory T cells, and tumor-fighting effector cells, which are closely related to the effect of cancer therapy. The tumors are categorized into cold or hot, which is dependent on the production of inflammatory cytokine and T cell infiltration [21]. Hot tumors exhibit a strong response to immunotherapy, while so-called cold tumors exhibit striking features of T cell absence or exclusion [22]. Thus, a large number of studies have focused on converting cold tumors to hot tumors to improve the therapeutic efficacy [21,23].

Inflammatory cytokines including TNF-α, IL-6 and IL-8 play a major role in cancer-associated immune response [24,25,26]. TNF-α is secreted by macrophages/monocytes during acute inflammation, which participates a variety of signaling events [27]. The anti-tumor activity of TNF-α has been fully proved, and it is considered to be involved in multiple mechanisms, such as the activation of anti-tumor immune responses and the promotion of cancer cell apoptosis [28,29]. IL-6 is a prototypical cytokine with functions like TNF-α [30], which plays the role of maintaining host homeostasis. Once homeostasis is disrupted by infection or tissue injuries, IL-6 is produced and contributes to host defense through the activation of immune responses [31,32]. IL-8 is secreted by phagocytes and mesenchymal cells when stimulated by inflammation, which activates neutrophils and then induces chemotaxis, exocytosis and the respiratory burst [33,34]. In this article, we firstly illustrated that the levels of TNF-α, IL-6 and IL-8 were significantly enhanced by the treatment of oncoVV-AVL. This spurred us to further explore the courses for the elevation of these cytokines.

Transcription factor nuclear factor-κB (NF-κB) is mainly responsible for regulating a series of cellular processes, such as immune and inflammatory reactions, cell growth, development and apoptosis [35]. It has been accepted as the master regulator of inflammation and immune homeostasis. Activator protein-1 (AP-1) is a dimeric transcription complex, involving in a variety of cellular physiological functions. AP-1 is implicated in various diseases, including cancer. As a dimer, AP-1 binds to homologous DNA sites within gene promoter elements to regulate the expression of a range of genes, affecting cell proliferation and differentiation [36]. The AP-1 family is composed of four sub-families: Maf(myofascial fibrosarcoma) (c-Maf, MafB, MafA, Mafg/f/k, Nrl), Fos(c-Fos, FosB, Fra1, Fra2), Jun(c-Jun. JunB, JunD), and ATF activated transcription factor (ATF2, LRF1/ATF3, DP1, JDP2) protein family [37]. Thought to be a transcriptionally active complex, AP-1 is primarily formed by Jun and Fos as a heterodimer, or by Jun alone to form a homodimer [38]. AP-1 activity is regulated by the MAPK pathway, as the phosphorylated MAPK/ERK binds with the promoter of c-Jun or c-Fos and rapidly induces their expression. Therefore, the activity of AP-1 mainly depends on its cellular environment and dimer composition, while the phosphorylation states of Fos and Jun significantly affect its activity [39]. Although AP-1 components can sometimes act as oncogene or tumor suppressor alone, they usually play a key role in upstream oncogenic events [40,41]. Both of AP-1 and NF-κB can induce the expression of a series of inflammatory factors, such as IL-6, IL-8 and TNF-α [33,42,43,44]. Once cells are stimulated by extracellular signals, they will initiate a series of inflammation-related signals, including NF-κB and AP-1, to response immediately. These molecules subsequently trigger a series of events such as transcription, translation, modification and subcellular localization, as well as protein stability [41,45]. Our results indicate that the expression of AP-1 and its constituent units c-Jun/p-c-Jun and c-Fos/p-c-Fos were also enhanced at protein levels after the treatment of oncoVV-AVL.

We further speculated that oncoVV-AVL infection stimulates the Ras/ERK pathway, which can be activated by stimulation such as growth factors and cytokines, then regulates the viral replication [46,47]. After activation, ERK acts as a serine/threonine protein kinase to phosphorylate c-Fos and c-Jun [48]. Subsequently, p-c-Fos combined with p-c-Jun to form AP-1 to trigger an inflammatory response. Once AP-1 is activated, the inflammatory response is initiated. After interacting with DNA, AP-1 can regulate the expression of upstream genes, influence cell proliferation and differentiation, and play a leading role in promoting and inhibiting tumorigenesis. Our results are in line with the previous study that found AVL promoted the activation of ERK [9]. Therefore, we preliminarily determine that the transcription factor AP-1 and its related signaling pathways may participate the process of oncoVV-AVL inhibiting tumor cells. Further studies are needed to explore the details. 

Our previous studies demonstrated that AVL gene insertion significantly enhanced the tumor-killing effect of oncoVV [9,10,11], while the relationship of oncoVV-AVL infection and immunity response was not investigated. The association of oncoVV-AVL and the activation of AP-1 provides a novel insight for the application of oncolytic vaccinia virus in tumor immunotherapy. At present, several researchers have studied the process of OVs converting a noninflamed tumor into an inflamed one. Intratumorally injection of mJX-594, a targeted and GM-CSF-armed oncolytic vaccinia virus, may increase the sensitivity to anti-PD-1 and/or anti-CTLA-4 therapy [49]. Our study provides a new OVs for the investigation of tumor-promoting players to elicit stronger immune defense to tumors. 

To summarize, our results provide a new approach to develop and utilize marine resources. In addition, a potentially therapeutic strategy to “heat up” cold tumors could lie in the application of an oncolytic virus as priming therapy.

## 4. Materials and Methods

### 4.1. Cell Culture

Human hepatocellular carcinoma cell lines including PLC/PRF/5, Hep3B, SK-HEP-1 and HuH-7 used in present study were obtained from Hangzhou Qiannuo Biotechnology Co., Ltd. (Hangzhou, China). All cells were cultured in DMEM medium (Gibco, Thermo Fisher Scientific, Waltham, MA, USA) supplied with 10% fetal bovine serum (Hyclone Laboratories) and 1% Penicillin-Streptomycin. The cells were incubated at 37 °C in a humidified atmosphere containing 5% CO_2_. 

### 4.2. Quantitative Real-Time PCR

The cells were planted into 24-well plates in triplicate, and then all cells were cultured for 12 hours. After that, the cells were infected by the viruses for 36 hours. Finally, the total RNA of cells was extracted and reverse transcribed into cDNA with a ReverTra Ace qPCR RT Kit (Toyobo, Japan). The cDNA was amplified utilizing SYBRR Green Realtime PCR Master Mix (Toyobo, Japan) according to the manufacturer’s instructions. The expression levels of each gene were calculated by 2^^-ΔΔCT^ and normalized by GAPDH. The primers used were followed as: IL-6, 5′-CAATCTGGATTCAATGAGGAGAC-3′ and 5′-TGTTCTGGAGGTAC TCTAGG TAT-3′; IL-8, 5′-GAGTGCTCTAGTATGTTGTGTCAA-3′ and 5′-CCAAAGCCA TTCT CTTACAGATTC-3′; TNF-α, 5′-CTAAGAGGGAGAGAAGCAACTAC-3′ and 5′-T CAGT ATGTGAGAGGAAGAGAAC-3′; GAPDH, 5′-GACAGTCAGCCGCATCTTCT-3′ and 5′-GCGCCCAATACGACCAAATC-3′.

### 4.3. ELISA Assay

After 36 hours of virus infection, the cell supernatants were collected and determined by ELISA according to the manufacturer’s instruction (Multi sciences Biotech, Hangzhou, China).

### 4.4. Dual Luciferase Reporter Gene Assay

Cells were planted into 96-well plates (8 × 10^3^/well) with 4 replicates. After 12 hours, the Renilla luciferase reporter vector PRL-TK and the plasmid with target gene (AP-1 or NF-kB) were co-transfected into the cells with a mass ratio of 1:500. PRL-TK served as the internal control. The next day, the cells were treated with PBS, oncoVV or oncoVV-AVL at an MOI of 5 for 36 h. Subsequently, the luciferase activities were determined with a chemiluminescence instrument by using a dual-luciferase assay kit (GeneCopoeia, Inc., Rockville, MD, USA). 

### 4.5. Western Blot

Cells were collected and lysed. Then SDS-PAGE electrophoresis was performed to separate proteins of different molecular weights. Subsequently, the protein samples were transferred to PVDF membranes. The membranes were incubated at 4 °C with primary antibodies (1:4000 dilution) overnight. Next, membranes were incubated with secondary antibodies (1:4000 dilution) for 1.5 h at room temperature. Finally, the protein bands on the membrane were scanned with a chemiluminescence image system (Clinx, Shanghai, China). The primary antibodies, including GAPDH, c-Jun, c-Fos, p-c-Jun and p-c-Fos, were purchased from Cell Signaling Technology. The corresponding secondary antibodies were HRP-conjugated goat anti-rabbit IgG (AS014, ABclonal, Wuhan, China). 

### 4.6. Data Analysis

For data analysis, GraphPad Prism 8.0 software (GraphPad Software, San Diego, CA, USA) was used. The one-way ANOVA test was performed for comparison among groups. All results were showed in mean ± SEM, and only *p* < 0.05 was considered to be statistically significant.

## 5. Conclusions

The limitation of the present research is the lack of isolated AVL control. Therefore, it is still impossible to conclude that oncoVV-AVL is superior to isolated AVL in the treatment of cancer. In summary, *Aphrocallistes vastus* lectin promotes the inflammatory response induced by oncolytic virus, thereby enhancing the anti-cancer effect of oncolytic virus in hepatocellular carcinoma, which may be associated with AP-1 activity. Given the role of oncoVV-AVL against cancer, the further investigation of immunity response stimulated by oncoVV-AVL is necessary. These results provide a novel prospect for the application and exploitation of *Aphrocallistes vastus* lectin.

## Figures and Tables

**Figure 1 marinedrugs-20-00667-f001:**
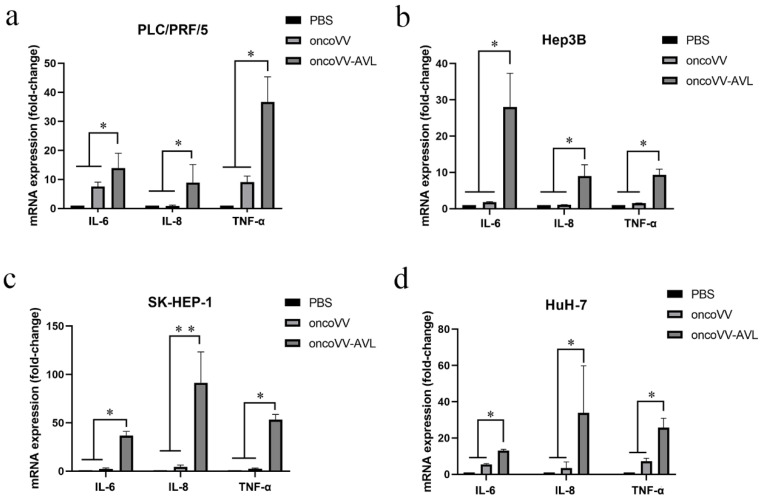
oncoVV-AVL increased the expression of inflammation cytokines at mRNA level. After 36 hours of infection with oncoVV or oncoVV-AVL in (**a**) PLC/PRF/5 cells, (**b**) Hep3B cells, (**c**) SK-HEP-1 cells, and (**d**) HuH-7 cells. The mRNA levels of IL-6, IL-8 and TNF-α were detected by qRT-PCR assay. Data were showed in mean ± SEM, and experiments were performed in triplicate. (* *p* < 0.05, ** *p* < 0.01).

**Figure 2 marinedrugs-20-00667-f002:**
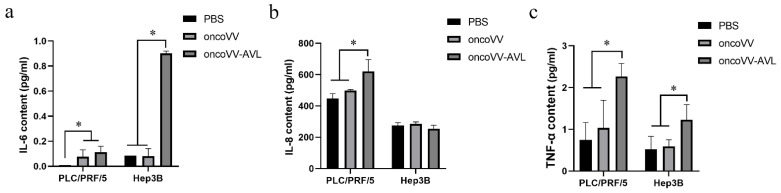
The contents of IL-6, IL-8 and TNF-α in cell supernatant of PLC/PRF/5 and Hep3B cells were measured by ELISA. (**a**) The content of IL-6. (**b**) The content of IL-8 (**c**) The content of TNF-α. Data were showed in mean ± SEM, and experiments were performed in triplicate. (* *p* < 0.05).

**Figure 3 marinedrugs-20-00667-f003:**
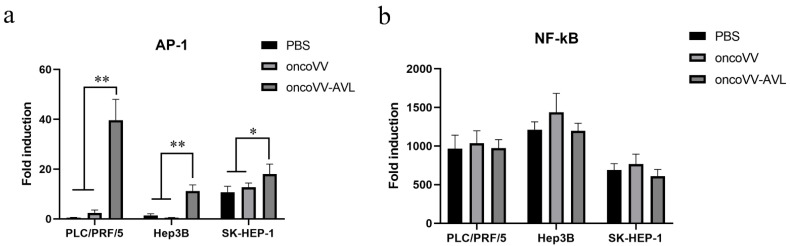
oncoVV-AVL significantly increased the transcriptional activity of AP-1 in HCC but not NF-κB in HCC. Transcriptional activity of AP-1 (**a**) and NF-κB (**b**) were detected in PLC/PRF/5, Hep3B and SK-HEP-1 cells after oncoVV or oncoVV-AVL infection, with PBS as a negative control. Data were showed in mean ± SEM, and experiments were performed in triplicate. (* *p* < 0.05, ** *p* < 0.01).

**Figure 4 marinedrugs-20-00667-f004:**
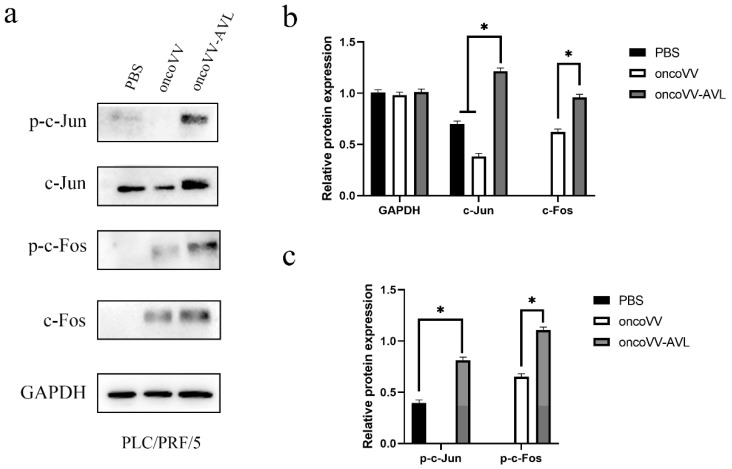
oncoVV-AVL significantly enhanced the production of c-Jun, c-Fos and their phosphorylation in PLC/PRF/5 cells. (**a**) The production of c-Fos and c-Jun proteins in PLC/PRF/5 cells. (**b**) Quantification of total c-Fos and c-Jun proteins. (**c**) Quantification of phosphorylated c-Fos and c-Jun proteins. Data were showed in mean ± SEM, and experiments were performed in triplicate. (* *p* < 0.05).

**Figure 5 marinedrugs-20-00667-f005:**
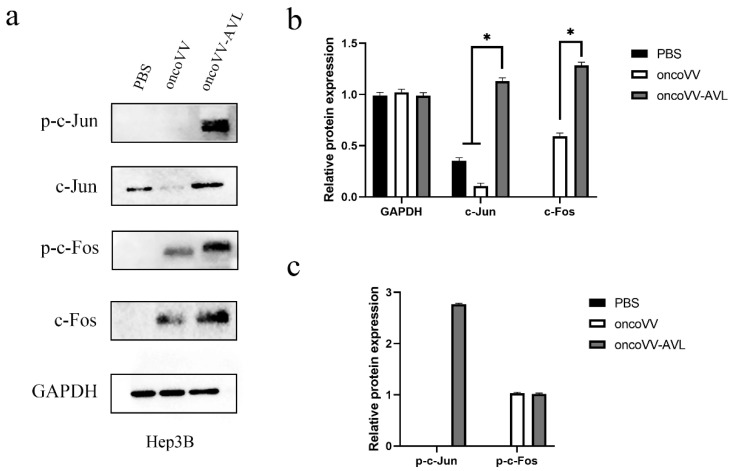
oncoVV-AVL significantly enhanced the production of c-Jun, c-Fos and their phosphorylation in Hep3B cells. GAPDH was used as the loading control, and PBS was used as the negative control. (**a**) The production of c-Fos and c-Jun proteins in Hep3B cells. (**b**) Quantification of total c-Fos and c-Jun proteins. (**c**) Quantification of phosphorylated c-Fos and c-Jun proteins. Data were showed in mean ± SEM, and experiments were performed in triplicate. (* *p* < 0.05).

**Figure 6 marinedrugs-20-00667-f006:**
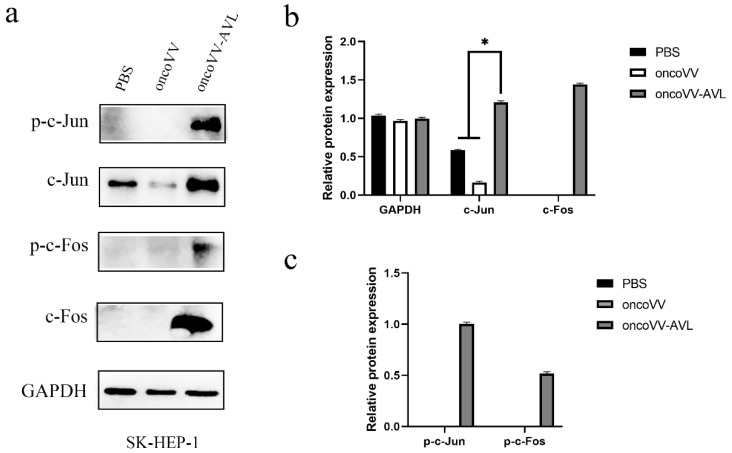
oncoVV-AVL significantly enhanced the production of c-Jun, c-Fos and their phosphorylation in SK-HEP-1 cells. GAPDH was used as the loading control, and PBS was used as the negative control. (**a**) The production of c-Fos and c-Jun proteins in SK-HEP-1 cells. (**b**) Quantification of total c-Fos and c-Jun proteins. (**c**) Quantification of phosphorylated c-Fos and c-Jun proteins. Data were showed in mean ± SEM, and experiments were performed in triplicate. (* *p* < 0.05).

**Figure 7 marinedrugs-20-00667-f007:**
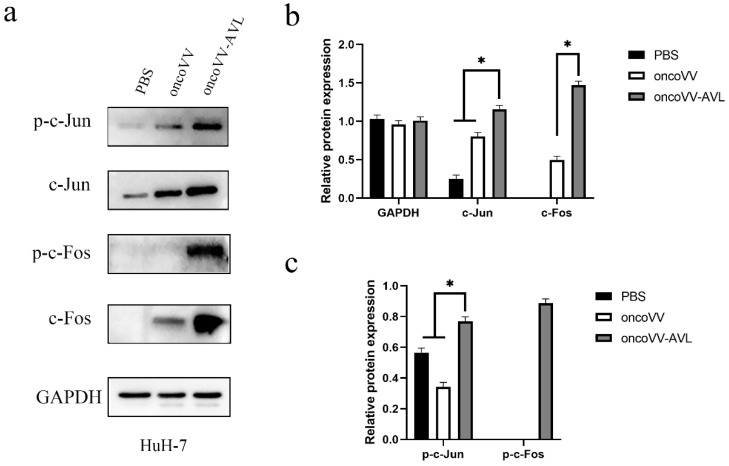
oncoVV-AVL significantly enhanced the production of c-Jun, c-Fos and their phosphorylation in HuH-7 cells. GAPDH was used as the loading control, and PBS was used as the negative control. (**a**) The production of c-Fos and c-Jun proteins in HuH-7 cells. (**b**) Quantification of total c-Fos and c-Jun proteins. (**c**) Quantification of phosphorylated c-Fos and c-Jun proteins. Data were showed in mean ± SEM, and experiments were performed in triplicate. (* *p* < 0.05).

## Data Availability

Data are contained within the article.

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
