# Peer review of "Oncolytic Vaccinia Virus Carrying Aphrocallistes vastus Lectin (oncoVV-AVL) Enhances Inflammatory Response in Hepatocellular Carcinoma Cells"

_marinedrugs, 2022, doi:10.3390/md20110667_

Round 1

Reviewer 1 Report

1. The introduction should be more generalized. This is the continuation of a previous work, but authors should rewrite it focusing on related common terms. The introduction should be comfortably followed by the readers with no previous idea on this research topic.

2. Results 2.1: The results are not explained in 2.1 as the authors did in 2.2. Please do so.  Based on cell types, expression of inflammatory cytokines varied a lot. Can the authors explain it?

3. Results 2.2: We can see that the results for PLC/PRF/5 (IL-6 content) and Hep (3B) (IL-8 content) is not very significant. So, the question comes - why did the authors exclude SK-HEP-1 and Huh-7 cells in this experiment? Please explain.

4. It will be better if the authors refer some more articles similar to this work and include those in the discussion. 

Reviewer 2 Report

The topic is of interest but presentation is poor. Manuscript needs a substantial editing/restructuring mostly to place discussion-like paragraphs into a discussion section from Results section. 

Major concern is about the lack of discussion about the role played by AVL lectin and lectins in general in a) stimulating immune cells, and b) killing tumor cells directly. There is no answer how expressed AVL lectin stimulates nuclear factor expression and activity in cancer cells and cytokine production. Does this happens due to intracellular action of expressed lectin or through extracellular stimulation of cancer cells? Study design would benefit by combining extracellular AVL lectin application with cell infection with oncoVV.

Authors talk about the stimulation of the immune response but study readouts are from cancer cells. This is a serious disconnect.

Statistical comparison must be done by ANOVA with p-values corrected for multiple comparison as current study has three variables (PBS, oncoVV, and oncoVV-AVL). The use of a t-test in this design is incorrect.

Most other comments and suggestions are made directly in the manuscript file as comments/highlights.

Reviewer 3 Report

The manuscript entitled “Oncolytic vaccinia virus carrying Aphrocallistes vastus lectin (oncoVV-AVL) enhances the host inflammatory response” by Zheng et al., describes the “association of oncoVV-AVL infection and the cellular immune response in hepatocellular carcinoma cells”. This work follows a successful research line of the group, which is focused on the study of the potential use of the vaccinia virus incorporating the A. vastus lectin as an oncolytic virus for cancer treatment.

            The work is scientifically solid and the manuscript is well written, as expected from the past experience of the authors on this system. Although the conclusions are well supported by the experimental results, the lack of controls in all the assays, using the AVL lectin alone does not permit to discriminate between the effects of the lectin itself from the effects of the incorporation of this lectin on the vaccinia virus.

            In this sense, authors indicated in a previous article that “However, due to isolated AVL being unavailable in our laboratory at present, we are unable to determine whether isolated AVL affects cancer cells with or without the presence of oncoVV. Furthermore, it is still impossible for us to draw a conclusion that oncoVV-AVL has advantage over isolated AVL in treating cancers” (Wu et al., Marine Drugs 2019, 17, 363; doi:10.3390/md17060363).

            In conclusion, this lack of necessary controls without isolated AVL compromises the claimed efficiency of oncoVV-AVL as a tool for cancer treatment and therefore the scientific relevance of this work. Authors should respond to this point and indicate the reason why they do not have isolated lectin.

            In this regard, important changes have occurred since 2019 in the structural biology realm: AlphaFold is now available to predict 3D protein structures accurately. As authors may know, the predicted 3D structure of the lectin is publicly available in UniProt since all proteins from this database are predicted. There are two entries for A. vastus lectin (Q9BIG7 and Q9BIG8) which exhibit 87% seq. identity. Most probably, this tool would help authors to produce recombinant lectin and this would make their studies more solid.  

Round 2

Reviewer 1 Report

Satisfied with the responses and the modified title.

Reviewer 3 Report

It can be deduced from the response of the authors that they are perfectly aware of the limitations of the conclusions that can be drawn from their experiments. Although strictly speaking this is not a valid response to my concern since the results from the control with the oncoVV-AVL can be compromised by the action of the lectin alone, I feel that the work deserves publication. Nevertheless, authors should indicate in conclusions that they are dealing with this issue and that their results should be taken with caution.
